# Efficacy of an Enterotoxigenic *Escherichia coli* (ETEC) Vaccine on the Incidence and Severity of Traveler’s Diarrhea (TD): Evaluation of Alternative Endpoints and a TD Severity Score

**DOI:** 10.3390/microorganisms11102414

**Published:** 2023-09-27

**Authors:** Nicole Maier, Shannon L. Grahek, Jane Halpern, Suzanne Restrepo, Felipe Troncoso, Janet Shimko, Olga Torres, Jaime Belkind-Gerson, David A. Sack, Ann-Mari Svennerholm, Björn Gustafsson, Björn Sjöstrand, Nils Carlin, A. Louis Bourgeois, Chad K. Porter

**Affiliations:** 1PATH, Washington, DC 20001, USA; lbourgeois@path.org; 2Johns Hopkins Bloomberg School of Public Health, Baltimore, MD 21205, USA; shannon.grahek@gmail.com (S.L.G.); jhalper2@jhu.edu (J.H.); susrtpo@gmail.com (S.R.); ftroncos@earthlink.net (F.T.); jcshimko@gmail.com (J.S.); dsack1@jhu.edu (D.A.S.); 3Laboratorio Diagnostico Molecular, Guatemala City 01009, Guatemala; otorres@dxmolecular.com; 4Children’s Hospital Colorado, Aurora, CO 80045, USA; jaime.belkind-gerson@childrenscolorado.org; 5Department of Microbiology and Immunology, University of Gothenburg, 405 30 Gothenburg, Sweden; ann-mari.svennerholm@microbio.gu.se; 6Scandinavian Biopharma Holding AB, 171 48 Stockholm, Swedenbjorn.sjostrand@scandinavianbiopharma.se (B.S.); nils.carlin@etvax.se (N.C.); 7Naval Medical Research Command, Silver Spring, MD 20910, USA; chad.k.porter2.civ@health.mil

**Keywords:** ETEC, enteric vaccine, traveler’s diarrhea, severity score

## Abstract

The efficacy of an Oral Whole Cell ETEC Vaccine (OEV) against Travelers’ Diarrhea (TD) was reexamined using novel outcome and immunologic measures. More specifically, a recently developed disease severity score and alternative clinical endpoints were evaluated as part of an initial validation effort to access the efficacy of a vaccine intervention for the first time in travelers to an ETEC endemic area. A randomized, double-blind, placebo-controlled trial followed travelers to Guatemala or Mexico up to 28 days after arrival in the country following vaccination (two doses two weeks apart) with an ETEC vaccine. Fecal samples were collected upon arrival, departure, and during TD for pathogen identification. Serum was collected in a subset of subjects to determine IgA cholera toxin B subunit (CTB) antibody titers upon their arrival in the country. The ETEC vaccine’s efficacy, utilizing a TD severity score and other alternative endpoints, including the relationship between antibody levels and TD risk, was assessed and compared to the per-protocol primary efficacy endpoint. A total of 1435 subjects completed 7–28 days of follow-up and had available data. Vaccine efficacy was higher against more severe (≥5 unformed stools/24 h) ETEC-attributable TD and when accounting for immunologic take (PE ≥ 50%; *p* < 0.05). The vaccine protected against less severe (3 and 4 unformed stools/24 h) ETEC-attributable TD when accounting for symptom severity or change in activity (PE = 76.3%, *p* = 0.01). Immunologic take of the vaccine was associated with a reduced risk of infection with ETEC and other enteric pathogens, and with lower TD severity. Clear efficacy was observed among vaccinees with a TD score of ≥4 or ≥5, regardless of immunologic take (PE = 72.0% and 79.0%, respectively, *p* ≤ 0.03). The vaccine reduced the incidence and severity of ETEC, and this warrants accelerated evaluation of the improved formulation (designated ETVAX), currently undergoing advanced field testing. Subjects with serum IgA titers to CTB had a lower risk of infection with ETEC and *Campylobacter jejuni/coli*. Furthermore, the TD severity score provided a more robust descriptor of disease severity and should be included as an endpoint in future studies.

## 1. Introduction

Travelers’ diarrhea (TD) is the most common travel-related illness, affecting approximately 10–14 million people annually [1,2,3,4]. Vaccine development remains a primary prevention strategy, particularly for the most common etiologies such as enterotoxigenic *Escherichia coli* (ETEC), *Shigella*, *Campylobacter*, and norovirus [4,5,6]. Additionally, the emergence of antimicrobial resistance (AMR) in common bacterial TD pathogens has increased the need for primary prevention and has led public health stakeholders to urge for the acceleration of vaccine development [7,8]. The use of clinically meaningful primary and secondary endpoints to assess disease is critical, since these endpoints can impact significantly on vaccine development investment, licensure, and uptake.

Outcomes to assess the efficacies of primary and secondary TD interventions (i.e., vaccines, passive immunoprophylaxis, and/or treatments) have historically been based on loose stool frequency with ≥1 associated gastrointestinal symptom. In addition, efficacy is often assessed based on a predominance of often mild TD and not on TD affecting traveler activities. Clinical practice guidelines for TD management have incorporated TD severity based on the functional impact on activity [9]. Additionally, recommendations from a Vaccines Against Shigella and ETEC (VASE) conference workshop highlight the need for disease severity scores for use in efficacy studies for TD primary and secondary preventions [10]. However, no published vaccine studies have incorporated functional impact into efficacy assessments [11,12,13,14,15,16,17,18].

Previous field studies of ETEC vaccines have shown that vaccination can impact disease severity; however, these assessments have primarily been based on the comparison of dichotomous clinical endpoints with associated signs and symptoms in vaccine and placebo groups, and they have lacked uniformity, harmonization, and consistency [15,16,17,18]. Consequently, efficacy estimates have varied. A composite disease score might improve the characterization of the breadth of TD severity and enable a more robust assessment of vaccine effect. For this reason, we recently developed a TD scoring system to better characterize disease severity independent of etiology, and to serve as a secondary outcome measure in future field trials [19]. Upon development and initial evaluation, the severity score better predicted a negative impact on activity than did any individual sign or symptom [19], confirming that a composite score may be a more informative endpoint to characterize TD illness than the utilization of a single disease parameter, such as the frequency of loose stools. Nevertheless, a further validation of this score is needed and its potential utility in assessing primary TD prevention interventions needs to be evaluated. To accomplish this, the TD score was recently applied retrospectively to a Phase 3 trial of an Oral Whole Cell ETEC Vaccine (OEV) expressing common ETEC colonization factor (CF) antigens (CFA/I, CS1, CS2, CS3, CS4, and CS5) and supplemented with recombinant cholera toxin B subunit (rCTB; 1 mg per dose) [18,20,21]. This trial was conducted in adults traveling from the United States to Guatemala and Mexico for Spanish language training.

OEV was previously shown to protect Austrian travelers against ETEC TD [13], and it also protected travelers to Guatemala and Mexico against more severe ETEC TD caused by ETEC strains sharing OEV antigens (PE = 77%, *p* = 0.03) [18]. The follow-on trial detailed here used a commercial scale formulation of the vaccine (OEV-118) with a different recombinant CTB, and again evaluated the vaccine’s efficacy against ETEC TD among travelers to Guatemala and Mexico [21]. We evaluated the impact of vaccine immune responses on vaccine efficacy and calculated the efficacy using novel considerations regarding the impact of the TD on activities as well as the novel disease severity score.

## 2. Materials and Methods

### 2.1. OEV-118 Phase 3 Study

This is a reanalysis of data from a Phase III double-blind, randomized, placebo-controlled vaccine efficacy field trial using alternative disease endpoints and the TD severity score, which captures a wider spectrum of illness severity as compared to the original study endpoints [21]. The OEV-118 field trial was conducted from 1999 to 2002 to evaluate the safety and efficacy of an oral vaccine prepared from formalin-killed whole cell ETEC strains expressing CFs CFA/I and CS1 through CS5, supplemented with 1 mg of recombinant B subunit of cholera toxin (rCTB) per dose [21]. Subjects received either the vaccine in a bicarbonate citrate buffer, or a placebo formulation of K12 *E. coli* devoid of CFs and CTB and similar in appearance to the vaccine in the same buffer. Vaccine was shipped to participants’ homes for immunization supervised by a study nurse over the phone but not directly observed. The trial used local travel clinics and laboratories in Guatemala and Mexico, which utilized the same sample collection, processing, and pathogen identification methods used in an earlier pilot trial of the same product [18,21,22]. In addition to the ETEC identification methods (toxin: heat stabile toxin, ST; health labile toxin, LT and CF expression), similar to those previously employed [18,21,22], laboratories in Guatemala and Mexico also participated in a monthly testing program throughout the trial, in which they were sent coded samples to test proficiency in pathogen identification. Both laboratories consistently scored 85–100% on these proficiency panels. The study was approved by the Joint Committee on Clinical Investigation at Johns Hopkins University (CPMS Protocol No. 238879/004 (ETEC-004/OEV-118); approved 18 June 1999).

### 2.2. Study Population

OEV-118 study volunteers (N = 1458) were recruited, provided consent, and completed the enrollment process while in the United States, prior to travel to Antigua, Guatemala, or Cuernavaca, Mexico. Eligibility criteria for the OEV-118 trial are presented in Appendix A. To be included in vaccine efficacy reanalysis, subjects had to consume 2 doses of study product (vaccine or placebo); travel 7 to 28 days after the 2nd dose; and submit stool samples for microbiological and parasite analysis as per protocol, and have symptom data available.

### 2.3. Health Data and Sample Collection

Health information was collected through study clinic visits and the completion of daily diary cards while traveling. Diaries were provided to travelers by a study nurse upon arrival and collected weekly at the study site. Diaries were reviewed with study staff weekly, at departure, and during all TD episodes. The data collected included information on gastrointestinal and systemic symptoms, and the impact on activity. All solicited symptoms collected on the diary cards are presented in Appendix A. TD was defined as ≥3 unformed stools in a 24-h period plus one of the following: abdominal pain/cramps, nausea, vomiting, urgency, excessive gas, and fever. Stools were characterized as in the previous Phase 3 study [18]. A diarrhea episode was considered complete on the last day after which 72 h passed symptom-free with no additional unformed stools. Other solicited symptoms were graded as being mild if they were noticed but did not impact on activity; moderate if they caused some limitation in activity; and severe if they impacted activity to the point of non-participation [18,21]. Stool samples were collected upon arrival, travel day 7, and departure or travel day 21 if not yet departed. Additional samples were collected during diarrheal episodes and prior to antibiotic treatment if needed. ORS was routinely provided to participants with TD. If IV fluids or more extensive supportive treatment were required, subjects were admitted to a local inpatient treatment facility. Stools were self-collected into plastic disposable containers. Stool samples were added to three vials containing either Cary Blair transport medium (CB media) for bacterial identification, phosphate buffered saline (PBS) for norovirus and rotavirus identification, or sodium acetate/acetic acid/formaldehyde (SAF) for parasite identification. These vials were transported to the travel clinic within 24 h of collection [18].

### 2.4. Alternative Endpoints and TD Score Calculation

To evaluate vaccine efficacy across the spectrum of ETEC illness severity, several alternative illness endpoints were also considered in addition to the original per-protocol primary endpoint. Additional secondary endpoints and the impact of immunologic take were evaluated as per the recommendations of the primary trial Data Safety Monitoring Board (DSMB), and to align more closely with WHO ETEC Vaccine Product Preferred Characteristics [4,5], as well as the endpoint definitions used in a recently completed study of Finnish travelers to Benin evaluating the reformulated OEV plus dmLT adjuvant (NCT03729219; EudraCT2016-002690-35). The DSMB recommendation regarding these post hoc analyses of vaccine efficacy were driven by concerns that the original ETEC case definition may have been too restrictive, leading to a limited number of ETEC outcomes for the primary efficacy analysis, and that the novel immunization approach used during the trial (home-based immunization, with dosing being supervised by a study nurse over the phone) may have contributed to a lower vaccine response than anticipated among vaccine recipients. A subset of trial participants (n = 646) had anti-CTB serum IgA titers evaluated upon arrival to Guatemala and Mexico, with the median arrival serum among vaccinees (n = 328) being 1358 [21]. Interestingly, approximately 75% of the vaccine-preventable outcome (VPO) cases in vaccinees based on the per-protocol definition had anti-CTB serum IgA titers below the median, indicating a possible level of ineffective immunization potentially associated with the at-home vaccination approach. Finally, vaccine efficacy was further evaluated, utilizing TD score thresholds, to estimate the reduction in disease severity in vaccine recipients versus placebos, including immunologic take versus non-take groups, and the overall vaccine impact assessed, compared to traditional endpoint analysis.

A summary of TD endpoint definitions, their abbreviations, and the rationale for inclusion in this reanalysis can be found in Table 1. The original trial primary endpoint was vaccine-preventable ETEC-attributable TD (VPO-ETEC TD), defined as TD caused by an ETEC strain expressing at least one antigen contained in the OEV (LT, CFA/I, CS1, CS2, CS3, CS4, or CS5), with no other enteric pathogen being isolated during the window of 24 h before to 72 h after illness onset. For this endpoint, TD was defined as ≥5 unformed stools in a 24-h period, accompanied by abdominal pain, cramps, nausea, or vomiting of any intensity. Based on recommendations by the DSMB, we evaluated additional endpoints as well as a TD severity score adapted from the Vesikari score traditionally used for rotavirus-attributable diarrhea in pediatric populations (Table 2). The modified Vesikari Diarrhea Score incorporated objective signs of TD illness (diarrhea frequency/duration, vomiting frequency/duration, dehydration, fever, hospitalization) and was only calculated for subjects with TD (≥3 unformed stools in a 24-h period, accompanied by ≥1 accompanying gastrointestinal (GI) symptom of any intensity/severity). Alternative endpoints included ETEC TD and more severe ETEC TD (MS-ETEC TD). ETEC TD was similar to VPO-ETEC TD, except that only ≥3 unformed stools in a 24-h period, accompanied by abdominal pain/cramps, nausea, and/or vomiting of any intensity were needed to qualify as a case, with ETEC as the sole pathogen. MS-ETEC TD was similar to ETEC TD, except that only moderate or severe gastrointestinal symptoms were considered, or that the illness impacted on activity. Two additional endpoints were included to align with the preferred product characteristics for ETEC vaccine candidates as presented at 2020 World Health Organization (WHO) Product Development for the Vaccines Advisory Committee (PDVAC) meeting [5]. These include Clinically Significant ETEC TD (CS ETEC TD) and More Severe Clinically Significant ETEC TD (MS-CS ETEC TD). CS ETEC TD was defined as ≥4 unformed stools in a 24-h period, accompanied by abdominal pain/cramps, nausea, and/or vomiting of any intensity, plus infection with ETEC sharing OEV antigens as the sole pathogen and isolated anytime during the diarrhea episode. More Severe Clinically Significant ETEC TD (MS-CS ETEC TD) was similar to CS ETEC TD, except that the gastrointestinal symptom must have been moderate or severe, or the illness impacted on activity. The restrictive time around ETEC isolation as part of the original per-protocol VPO definition (24 h before to 72 h after illness onset; see Table 1) significantly limited the number of ETEC outcomes for analysis. Nevertheless, these secondary definitions result in additional ETEC cases being included in the analysis of an immunologic take on OEV efficacy and enabled alignment with endpoint definitions of the reformulated OEV vaccine plus a dmLT adjuvant that has recently completed efficacy evaluation in a study of Finnish travelers to Benin (OEV-123) [5,20]. A TD score was also calculated for all participants, utilizing a 3-component TD severity score, as previously described [19].

### 2.5. Pathogen Determination

Collection, transport, and stool sample analysis were the same as previously described [18,22,23,24,25]. In brief, samples in Cary Blair transport media and PBS were stored at 2–8 °C, and samples in SAF were stored at room temperature. All samples were shipped temperature-controlled from the study clinics via courier within 24 h of submission to the corresponding laboratory, either the Instituto de Nutrición de Centro América y Panamá laboratory (INCAP) (Guatemala City, Guatemala) or the Hospital del Nino Morelense laboratory (Cuernavaca, Mexico). Samples were screened for pathogenic *E. coli*, *Salmonella* spp., *Shigella* spp., *Campylobacter jejuni*, and *Vibrio* spp. using standard bacteriological and immunological typing methods, selective media, and API [18,22,23,24,25]. *Giardia*, *Cryptosporidium*, *Cyclospora*, *Microsporidia*, and *Entamoeba histolytica* screening was also carried out using standard microscopic techniques, with suspected *Giardia* being confirmed through enzyme immunoassay. Norovirus and rotavirus screening were completed on samples collected for VPOs at CDC, Atlanta, USA; and on VPO samples and Non-VPO samples at the Johns Hopkins Bloomberg School of Public Health, using standard virology procedures [24].

Lactose-positive *E. coli*-like colonies were isolated on MacConkey agar for ETEC characterization, and five colonies were picked and preserved in 10% glycerol agar slants and 10% glycerol broth. GM1 enzyme-linked immunosorbent assay (GM-1 ELISA) was carried out to detect LT and ST, with CF determination for ETEC toxin-positive isolates being completed using dot blot assays [18,22,25].

### 2.6. Serum IgA CTB Antibody Determinations

Previous field studies of CTB or LT containing vaccines have suggested that immunization with these components may impact more generally on a participant’s apparent risk of infection and disease with other bacterial pathogens [11,15,26]. Blood samples from 646 subjects were available and evaluable for anti-CTB serum IgA antibody titer determination, pre-travel and/or upon arrival at the trial site. IgG responses were not assessed. The ELISA method used for these determinations has previously been validated and described [18,21]. We utilized an arrival titer of >360 to define subjects with presumptive immunologic take, to evaluate additional exploratory endpoints. Prior studies had shown that arrival titers in travelers above this level of anti-CTB antibody were associated with a significantly reduced risk of developing moderate to severe ETEC TD (~85%) [20].

### 2.7. Data Analysis

This is a reanalysis of a prospective double-blind, randomized, placebo-controlled Phase III vaccine efficacy trial (OEV-118) using active surveillance methods. All analyses were performed using SPSS or EpiInfo. All tests of hypotheses were one-sided and assessed at the 5% significance level. Protective efficacy was determined as [(AR_p_ − AR_v_)/AR_p_] × 100, where AR_p_ was the attack rate for placebo recipients, and AR_v_ was the attack rate for vaccine recipients. All *p*-values and confidence intervals were obtained from EpiInfo.

## 3. Results

### 3.1. Primary OEV Trial Analysis Using Per-Protocol VPO-ETEC TD Case Definitions

Among the 1458 participants vaccinated with OEV or placebo in the OEV-118 trial, 1435 completed symptom diary cards as well as 7–28 days of in-country surveillance. As in the previous field trial [18,21], OEV was extremely well tolerated (Appendix A). Overall vomiting post-dosing was rare but was associated with vaccination after both dose 1 and dose 2 (Appendix A). Two SAEs were reported during the OEV-118 trial, with neither being judged to be associated with the vaccine (Appendix A). The classic TD rate was 34% and 23% in Guatemala and Mexico, respectively, with approximately half meeting the definition of MS-ETEC TD (Table 1 and Table 3). There was no difference in the proportion of vaccine or placebo recipients meeting the per protocol VPO-ETEC TD primary endpoint (Table 3). Nevertheless, fewer OEV recipients (6/705, 0.85%) than placebo recipients (14/701, 2.00%) developed MS-ETEC TD; however, this was not statistically significant (PE = 57.4%, *p* = 0.06), possibly due to the low number of subjects meeting this endpoint. The most common ETEC outcome (~35%) was infection with ST strains expressing CS6, which were not covered by the OEV formulation under evaluation [22].

### 3.2. Post Hoc Assessment of Vaccine Efficacy: Impact of Immunologic Take and Application of Alternative Endpoints

There was significant protection against MS-ETEC TD (PE = 76.3%, *p* = 0.01) and MS-CS ETEC TD (PE = 80.5%, *p* = 0.01) among immunologic take subjects (arrival serum IgA anti-CTB titer > 360) (Table 4). No significant vaccine-induced protection was noted in immunologic non-take subjects against ETEC TD, even when moderate to severe GI symptoms or a change in activity were considered. The proportion of placebo recipients and vaccinees without immunologic take meeting these endpoints was comparable with efficacy estimates ranging from 3.1% to −70% (Table 4 legend). In contrast, vaccine recipients with immunologic take had reduced rates of all endpoints evaluated (PE 46.7–80.5%; Table 4).

### 3.3. TD Score Analysis

Figure 1 shows the distribution of the TD score across all 1435 participants in the study who had symptom data available for analysis; 412 participants met the classic definition of TD.

All subjects (100%) with a TD score ≥ 7 reported a negative impact to activity, with 60–90% of participants with a TD score of 3–6 changing their activity due to illness (Figure 2a). Among all participants in OEV-118, the odds of reporting a negative impact on activity increased with increasing TD score (OR = 2.68, *p* = 0.001) (Figure 2b).

Increasing protective efficacy was observed with increasing TD severity scores when vaccine recipients with immunologic take were compared to placebos (TD severity scores of ≥3 (PE = 63.8%; *p* = 0.05), ≥4 (PE = 81.8%; *p* = 0.01), and ≥5 (PE = 85.9%; *p* = 0.02)) (Table 5). Protection was also observed in the serology subset with increasing TD scores of ≥4 or ≥5, regardless of immunologic take (PE = 41.0% and 54.1%, respectively) (see legend of Table 5). Statistical significance was likely not achieved with increasing TD scores because of a limited number of outcomes among subjects with non-take in this analysis compared to placebo recipients (4, 2, and 1 participant achieving a TD score of >3, 4, or 5, respectively; while among placebo recipients, these outcomes were seen in 14, 14, and 9 participants, respectively). Significant vaccine protection was observed against TD of any etiology when TD scores of ≥5 (PE = 48.7%) or ≥6 (PE = 57.6%) were utilized and immunologic take was considered (Table 6).

Analysis of OEV vaccine protective efficacy in the full traveler set of 1435 was examined against the ETEC TD endpoint recommended by the trial’s DSMB (Table 1). In this TD score analysis, increasing protective efficacy was observed across the range of TD scores evaluated when all vaccines were compared to placebos (TD severity scores of ≥3 (PE = 45%; *p* = 0.08), ≥4 (PE = 72%; *p* = 0.01), and ≥5 (PE = 79%; *p* = 0.01) (Table 7). Vaccine protective efficacy in all vaccine recipients compared to placebo recipients with TD scores of ≥4 (PE = 83.5%; *p* = 0.006) or ≥5 (PE = 87.7%; *p* = 0.02) were higher against more severe ETEC (MS-ETEC VPO (Table 7).

### 3.4. Impact of Immune Status on Risk of Infection

The prospective design of the OEV-118 trial enabled an assessment of vaccine impact on the risk of symptomatic and asymptomatic infection with ETEC and other enteric pathogens. This analysis was initially carried out in the subset of subjects with serological data, given the assumption that an immune response to the vaccine would affect the risk of infection with ETEC and possibly other enteric pathogens. Of the 646 participants, 263 had arrival anti-CTB titer >360 (251 vaccine recipients and 12 placebos). Three hundred and eighty-three participants (77 vaccine recipients and 306 placebos) had anti-CTB titers of ≤360 upon their arrival in Guatemala or Mexico. When vaccine recipients with immunologic take (n = 251) were compared to all placebo recipients tested for CTB responses (n = 318), the risk of infection with an ETEC strain sharing an antigen with the vaccine was reduced by ~74% (RR = 0.26; *p* = 0.002) (Table 8).

Participants with arrival titers of >360 were also at reduced risk of these initial ETEC infections progressing toward either ETEC TD or more clinically significant MS-ETEC TD (*p* < 0.03 for all comparisons) (Table 8). Participants with arrival titers of >360 were also at reduced risk of these initial ETEC infections progressing toward either ETEC TD or the more clinically significant MS-ETEC TD (*p* < 0.03 for all comparisons) (Table 8).

An immunologic response was also associated with a significant reduction in the risk of *Campylobacter* infection (~66% reduction in risk) (RR = 0.34; *p* = 0.03, Table 9), but this did not seem to reduce campylobacteriosis severity. However, an impact on the progression to diarrheal illness was seen if a higher arrival titer of >640 was used as the threshold. Interestingly, there was also a trend toward risk reduction for Salmonella infection (RR = 0.32; *p* = 0.07) but not Shigella infection in those with an immunologic response to the rCTB of the vaccine (Table 9). Rotavirus infection was rare among travelers in this study (only a single infection being identified), while norovirus infection was more common [24]. Infections with rotavirus were not detected in VPO cases; norovirus cases (symptomatic and asymptomatic) were included if detected. 

## 4. Discussion

We reexamined the protective efficacy of a precursor to the current ETVAX [4,5,20,27,28] vaccine (OEV) against ETEC TD to assess the impact of disease severity and the immunological measures of effective immunization on vaccine efficacy. Specifically, we evaluated TD severity using a new disease scoring algorithm, and alternative clinical endpoints, as a first-round validation effort to assess the efficacy of a primary preventive intervention in a different traveler population, and in anticipation of its potential application to a recently completed ETVAX vaccine field efficacy study (OEV-123). We also evaluated the association between the effective ETEC vaccine immunization, as assessed via an anti-CTB response, and the risk of infection with ETEC and other bacterial enteropathogens. It should be noted that prior ETEC vaccine studies have reduced antibiotic use in field studies; however, these studies did not evaluate their ability to reduce the risk of infection [15,28].

The OEV-118 trial relied on dosing that was not directly observed. Rather, participants were dosed while on the telephone with a study nurse. As mentioned previously, the immunization process in the home setting may not have been effective at inducing serum anti-CTB IgA responses to vaccine components compared to immunization in a clinical setting [18,20,21]. Upon a review of the primary trial results, this became a focal point for the DSMB, who suggested that a reanalysis be conducted to assess the impact of immunological take as a correlation of successful immunization on vaccine efficacy. Reflective of the DSMB concern, anti-CTB response rates were lower among VPO cases in the vaccine recipients, suggesting that the immunologic take rates may have been more variable than in prior studies using directly observed vaccination [21,29].

Previous studies have suggested that serum anti-CTB IgA titers post-vaccination may serve as a simple marker for ETEC vaccine efficacy in the field [4,20]. We observed higher efficacy estimates among participants in the immunologic take group, compared to the non-immunologic take group, across all endpoint definitions and with a utilization of the TD severity score. These encouraging results need to be confirmed in subsequent travelers studies.

Importantly, participants (including vaccine and placebo recipients) arriving at the travel location with anti-CTB serum IgA titers above the threshold level of 360 were less likely to develop ETEC TD [20] associated with strains sharing an antigen with OEV, and experienced fewer infections with ETEC strains sharing a vaccine antigen (Table 8). Furthermore, given that asymptomatic infections can induce intestinal inflammation [4,30], vaccination by limiting infection may further reduce the risk of developing functional bowel disorders, which have been identified as long-term sequelae in 10–14% of travelers with TD. This observation was also important since significant reductions in infection and progression to disease suggest that the vaccine could further reduce the need for antibiotic intervention. In addition, it also suggests that vaccinated subjects may be less likely to acquire AMR strains during travel and disseminate them upon return to their country of origin.

Unexpectedly, participants with arrival titers above the 360-titer threshold were at a reduced risk of becoming infected with *Campylobacter* and *Salmonella* (Table 9). This reduction in risk did not carry over to ETEC strains that did not share an antigen with the vaccine, suggesting a somewhat selective reduction in risk. The observed reduction in the risk of infection with ETEC matching the vaccine, as well as *Campylobacter* and *Salmonella*, raises the question as to whether shared antigens yielded this reduced risk, or whether this was due to a non-specific effect triggered by the oral immunization process in the gut mucosa. It may also have been observed purely due to chance in this small study. Previous studies have shown that immunization with CTB- or LT-containing vaccines may impact on enteric illnesses due to other enteric pathogens such as *Campylobacter* and *Salmonella* [11,15], while immunization with the oral polio vaccine OPV has also been shown to reduce the risk of *Campylobacter* infection [31]. Albert and colleagues identified epitopes on CTB that cross-react with *Campylobacter* surface proteins, but these epitopes do not appear to be shared by LT [32]. Investigators from Zambia report an observation where immunization with ETVAX resulted in increased levels of antibodies that could cross-react against other diarrheagenic *E. coli* and several *Salmonella* virulence proteins [33]. It has been previously proposed that LT toxin exposure may make one more susceptible to infection with other enteric pathogens [26]. Vaccines such as OEV that induce anti-LT antibodies may neutralize this effect and reduce the risk of infection with other bacterial enteropathogens. More research is needed to confirm these interesting observations with OEV, and to better understand the associated mechanism(s).

OEV efficacy estimates were dependent on the TD endpoint definition used, as well as the strength of immunologic take. As in previous studies [18], OEV efficacy increased when more severe ETEC TD (≥5 unformed stools/24 h) was considered, particularly if immunologic take was included (PE range ≥ 50%, *p* < 0.05). Furthermore, the vaccine was protective against ETEC TD endpoints with reduced stooling frequency (≥3 or ≥4 unformed stools/24 h) when symptom severity or a change in activity were also factored into the definition. Given that these illnesses are most likely to interfere with travel plans, this vaccine may prove to be a highly useful travel medicine product. These results also underscore the need for alternative endpoints that may help to capture the spectrum of clinically relevant illnesses in an *a priori* manner.

The original protocol considered a modified scoring system for assessing TD severity, adapted from the Vesikari score traditionally used for rotavirus. This score was comparable between vaccine recipients and placebos in all analysis. One shortcoming of the modified Vesikari score is that it only included objective signs, with no consideration of the more subjective symptoms that have a demonstrable negative effect on participant wellness and activity level. Furthermore, the Vesikari score was originally developed to assess the severity of rotavirus in pediatric populations, and it may not include the signs and symptoms that are most relevant to an adult traveler population. We utilized a TD severity score specifically developed from an adult travel population [19], and thus, more directly applicable to the OEV-118 population. This severity score effectively accounted for the functional effect of TD, congruent with current guidelines on illness management [9]. Increasing protective efficacy was observed across the range of TD scores when limiting vaccine recipients to those with a clear immune response to the vaccine. Interestingly, clear efficacy was also observed against TD scores of ≥4 or ≥5, regardless of the immunologic response, an observation that was not seen when using the more traditional endpoints. Collectively, these data suggest that the TD score may be a more sensitive indicator of the overall vaccine impact. It is also important to note that based on the TD severity score, the vaccine was protective against higher TD scores, independent of etiology in those subjects with a clear immune response to the vaccine (a PE range of 48.7 to 57.6 for scores of ≥5 or 6, respectively; see Table 6).

A portion of this reanalysis was conducted in a population subset from which immunologic take was assessed. However, an overall vaccine take rate could not be determined, due the fact that a baseline serum sample could not be obtained prior to the first dose; rather, immunologic take was established from serum obtained from participants after dosing and upon arrival in the country. Another limitation was that only IgA responses to serum anti-CTB titers were assessed without an evaluation of anti-CTB IgG titers, immune responses to the colonization factors on which the vaccine is based, nor any other host factors that may have contributed to lower response frequencies. Follow-on studies with the next-generation formulation ETVAX vaccine should include comprehensive immunological evaluations against all vaccine components (colonization factors, LTB) and in all vaccinated participants, to address these limitations.

In summary, vaccination with OEV, as demonstrated by immunologic take, reduced the incidence of more severe ETEC in the cases analyzed. Prior analyses have demonstrated the importance of well-established ETEC case definitions and a consideration of immunologic take in assessing the vaccine efficacy [18,20]. Immunologic take was also an important marker for protection against infection with ETEC-containing antigen(s) shared with the vaccine, and surprisingly, also for protection against *Campylobacter* infection. Both observations deserve further exploration in future field trials, given their potential public health significance. In addition, a consideration of more severe disease and disease severity on the impact on activity should be incorporated into efficacy endpoints. Our results provided the first evidence indicating that the TD score may be a valuable new and possibly more sensitive metric for estimating vaccine efficacy, not only for ETEC, but for etiology-independent TD. While more traditional endpoint analyses are important, the disease score described here seems to provide a more sensitive and robust descriptor of disease severity that also considers the negative impact on activity. We propose that the TD severity score should be included as a secondary endpoint in future studies, to allow for further assessment and validation.

## Figures and Tables

**Figure 1 microorganisms-11-02414-f001:**
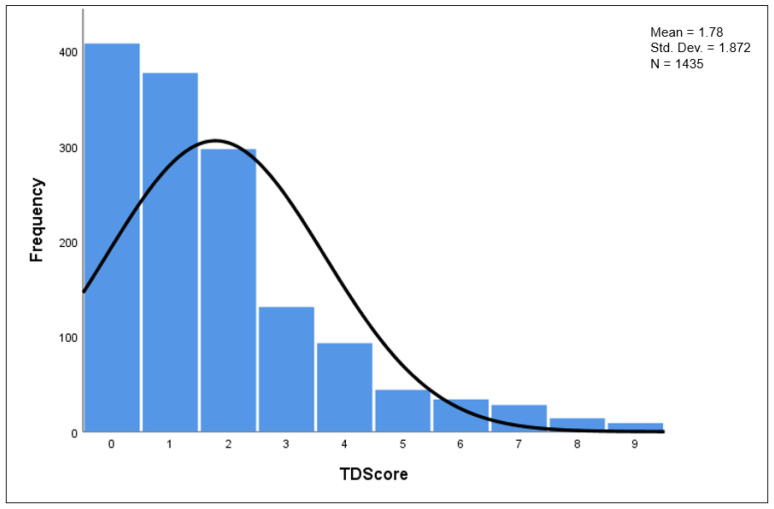
OEV-118 TD Score Distribution in Subjects Traveling to Guatemala or Mexico (N = 1435).

**Figure 2 microorganisms-11-02414-f002:**
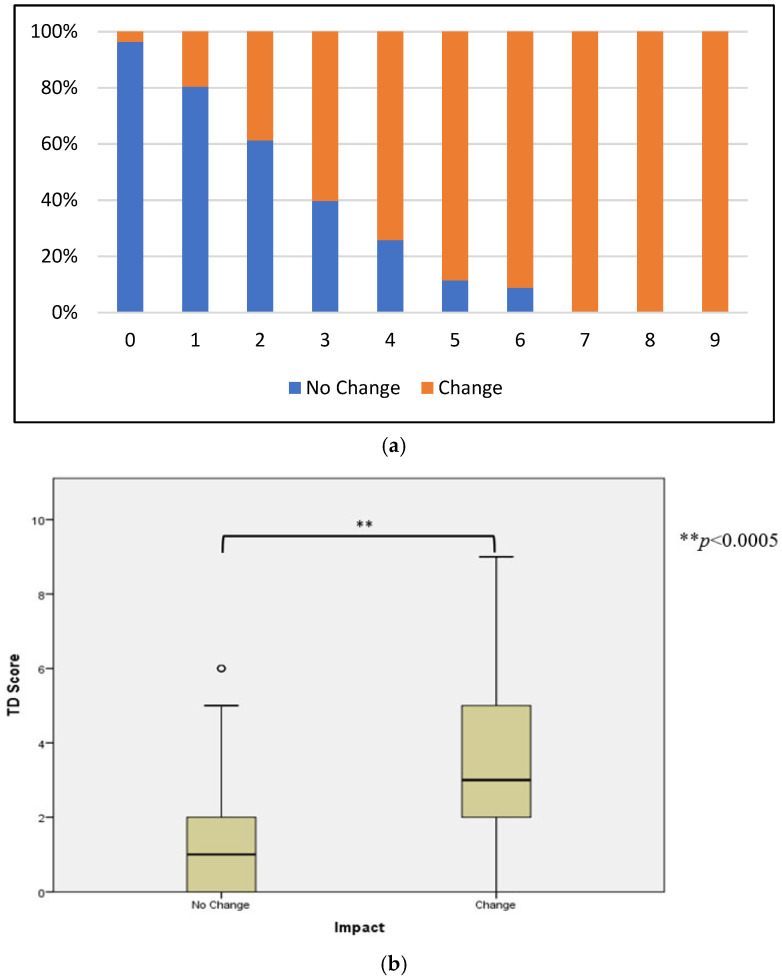
TD Severity Score and Impact on Activity. (**a**) Distribution of TD scores across OEV-118 subjects reporting change vs. no change in activity. (**b**) Logistic regression predicting likelihood of changing activity based on TD severity score.

**Table 1 microorganisms-11-02414-t001:** Travelers’ Diarrhea (TD) Endpoints.

Endpoint	Abbreviation	Definition	Rationale for Inclusion
Vaccine Preventable ETEC TD	VPO-ETEC TD	≥5 unformed stools in a 24-h period, accompanied by abdominal pain/cramps, nausea, and/or vomiting of any intensity; plus ETEC sharing antigens with OEV as the sole pathogen ^2^ and isolated in a window of 24 h before to 72 h after illness onset among subjects completing 2-dose regimen, traveling during the window of 7 to 28 days post 2nd dose, and completing 14 to 28 days surveillance	Original Study Endpoint
Classic TD	Classic TD	≥3 unformed stools in a 24-h period, accompanied by ≥1 accompanying GI symptom (abdominal pain or cramps, nausea, vomiting)	Classic TD Endpoint
ETEC TD	ETEC TD	≥3 unformed stools in a 24-h period, accompanied by abdominal pain or cramps, nausea, or vomiting of any intensity, plus ETEC as the sole pathogen ^2^ isolated	Recommended by DSMB
More Severe ETEC TD	MS-ETEC TD	ETEC TD, plus at least one moderate to severe GI symptom or ETEC TD, or changes in activity due to illness severity	Recommended by DSMB
Clinically Significant ETEC TD	CS ETEC TD	≥4 unformed stools in a 24-h period, accompanied by ≥1 abdominal pain or cramps, nausea, or vomiting of any intensity, plus ETEC as the sole pathogen ^2^ isolated	Developed to align with efficacy endpoints associated with the evaluation of reformulated OKV plus dmLT adjuvant (OEV-123) (NCT03729219); also aligns with preferred product characteristics for ETEC candidate vaccines presented at WHO 2020 PDVAC meeting ^1^
More Severe Clinically Significant ETEC TD	MS-CS ETEC TD	Clinically significant ETEC TD; plus at least one moderate to severe GI symptom or moderate-to-severe ETEC TD, or change in activity due to illness severity	Developed to align with efficacy endpoints associated with the evaluation of reformulated OKV plus dmLT adjuvant (OEV-123)

Abbreviations: TD = Traveler’s diarrhea; GI = gastrointestinal; DSMB = Data Safety Monitoring Board; dmLT = double mutant heat labile toxin; PDVAC = WHO’s Product Development Advisory Committee; OEV = Oral Inactivated ETEC Vaccine; CS = clinically significant; MS = more severe. ^1^ https://www.who.int/news-room/events/detail/2020/04/22/default-calendar/pdvac-2020, accessed on 20 June 2023. ^2^ Infections with rotavirus were not detected in VPO cases; norovirus cases (symptomatic and asymptomatic) were included if detected.

**Table 2 microorganisms-11-02414-t002:** Per Protocol Scoring System* for Estimating the Severity of the Illness.

Duration of Diarrhea	Points
Less than 2 days	1
2–3 days	2
4 or more days	3
Maximum number of diarrhea stools in 24 h	
3	1
4–5	2
More than 5	3
Duration of vomiting	
No vomiting	0
1–2 days	2
3 or more days	3
Maximum number of vomiting episodes in 24 h	
1	1
2	2
>2	3
Dehydration documented by provider	
None	0
Clinically present	2
Fever (oral) documented by coordinator or provider	
Less than 38 °C	0
≥38 °C	2
Hospitalization needed	
Yes	3
No	0

* Adapted from the Vesikari scoring system for rotavirus diarrhea.

**Table 3 microorganisms-11-02414-t003:** Point Estimates of OEV Protective Efficacy (PE) Against Vaccine-Preventable ETEC ^1^ and Diarrhea Severity via Various Endpoint Definitions.

Endpoint Definition ^1^	Vaccinees (N = 705)	Placebos (N = 701)	PE (%) (95% CI; *p*-Value)
VPO-ETEC TD (n = 13)	8	5	−63% (−91.1–79.0%; *p* = 0.29)
Modified Vesikari Diarrhea Score (median) ^2^	6.5	6	NS
ETEC-TD (n = 28)	11	17	35.7% (−36.6–69.0%; *p* = 0.17)
Modified Vesikari Diarrhea Score (median) ^2^	5	6	NS
MS-ETEC TD ^3^ (n = 20)	6	14	57.4% (−10.3–83.5%; *p* = 0.06)
Modified Vesikari Diarrhea Score (median) ^2^	5	6.5	NS

^1^ For all endpoint definitions, the ETEC isolate had to be obtained in the 24 h before to 72 h after TD onset. ^2^ Modified Vesikari Diarrhea Score, based on objective signs see Table 2), designed to compare severity of diarrheal illness in vaccine recipients and placebo recipients. Only developed for subjects meeting protocol definition of TD (>3 unformed or watery stools in 24-h period, accompanied by >1 accompanying gastrointestinal (GI) symptom (GI) of any intensity/severity (see Table 1). ^3^ Comparable PE (57%; *p* = 0.22) was seen if subjects indicated their ETEC TD episode caused a change in their daily activity. NS = Not Significant.

**Table 4 microorganisms-11-02414-t004:** Vaccine Protective Efficacy Against ETEC Diarrheal Illness Endpoints in Subset of subjects with arrival anti-CTB serum IgA titers: Vaccine Recipients ^1^ with OKV “Take” (n = 251) or Non-Take (n = 77) are compared to Placebo Recipients (n = 318).

	Protective Efficacy (95% CI; *p*-Value)			
Endpoint ^2^	ETEC TD	MS-ETEC TD ^†^	CS TD	MS-CS TD ^†^
Take (n = 251)	46.7% (−19.8–76.3%; *p* = 0.09)	76.3% (19.3–93.0%; *p* = 0.01)	54.8% (−23.6–83.5%; *p* = 0.09)	80.5% (19.4–98.6%; *p* = 0.01)

^1^ Vaccine Recipients with serum IgA titers against CTB of ≥360 upon arrival to their country of travel. ^2^ Details of endpoint definitions, available in Table 2. ^†^ Endpoint definition plus ≥1 moderate or severe GI symptom or illness of severity to cause change in activity. Note: Among the 77 vaccine recipients with arrival anti-CTB serum IgA titers of <360, no significant vaccine protective effect was seen for the endpoint included in the above Table. Attack rate for these endpoints in placebo and vaccine recipients without “Take” were comparable (PE ranged from 3.1% to −70%; *p* > 0.05 for all comparisons).

**Table 5 microorganisms-11-02414-t005:** Vaccine Protective Efficacy Against ETEC TD ^2^ and TD Severity Score in a Subset of Subjects with arrival anti-CTB serum IgA titers: Vaccine Recipients with OEV “Take” (n = 251) or Non-Take (n = 77) are compared to Placebo Recipients (n = 318).

	Protective Efficacy (95% CI; *p*-Value)		
	TD Severity Score ≥ 3	TD Severity Score ≥ 4	TD Severity Score ≥ 5
Take ^1^ (n = 251)	63.8% (8.6–87.9%; *p* = 0.05)	81.8% (21.1–95.8%; *p* = 0.01)	85.9% (10.5–98.6%; *p* = 0.02)

^1^ Vaccine Recipients with serum IgA titers against CTB of ≥360 upon arrival to their country of travel. ^2^ Details of endpoint definitions, available in Table 2. Note: Among the 77 vaccine recipients with arrival anti-CTB serum IgA titers of <360, there was a trend toward increasing vaccine efficacy against increasing TD severity scores compared to placebo recipients (n = 318). Protective efficacy for scores of >3, 4, or 5 were 15.5%, 41%, and 54.1%, respectively, but they did not achieve statistical significance (*p* > 0.05) due to a limited number of outcomes or sample sizes for the vaccine group.

**Table 6 microorganisms-11-02414-t006:** Vaccine Protective Efficacy Against TD Severity Score Independent of Etiology in a Subset of Subjects with arrival anti-CTB serum IgA titers: Vaccine Recipients ^1^ with OEV “Take” (n = 251) or Non-Take (n = 77) are compared to Placebo Recipients (n = 318).

Efficacy Estimate (95% CI)	Outcome	Outcome Description
	Etiology Independent	Serology subset population, independent of etiology and symptom severity.
15.4% (−7.9–33.8%)	Etiology Independent + TD severity ≥ 3	ETEC VPO (alt) and TD severity score ≥ 3
24.8% (−5.0–46.0%)	Etiology Independent + TD severity ≥ 4	ETEC VPO (alt) and TD severity score ≥ 4
48.7% (12.8–69.8%)	Etiology Independent + TD severity ≥ 5	ETEC VPO (alt) and TD severity score ≥ 5
57.6% (10.9–79.8%)	Etiology Independent + TD severity ≥ 6	ETEC VPO (alt) and TD severity score ≥ 6

^1^ Immunological “Take” among vaccine recipients was defined as anti-CTB serum IgA titer > 2560 in arrival sample.

**Table 7 microorganisms-11-02414-t007:** Vaccine Protective Efficacy Against Vaccine-Preventable ETEC via Various Endpoint Definitions in Study Population (N = 1435).

Efficacy Estimate (95% CI)	Outcome	Outcome Description
−59% (−38.4–48%)	ETEC VPO (per protocol)	≥5 unformed stools in 24 h, with abdominal pain/cramps, nausea, or vomiting, plus ETEC detected in limited time window
35.7% (−36.6–69.0%)	ETEC VPO (per protocol)	≥3 loose stools in 24 h with abdominal pain/cramps, nausea, or vomiting, plus ETEC detected at any time during disease
45% (−30.0–76.0%)	ETEC VPO + TD severity ≥ 3	ETEC VPO (alt) and TD severity score ≥ 3
72% (16.0–91.0%)	ETEC VPO + TD severity ≥ 4	ETEC VPO (alt) and TD severity score ≥ 4
79% (1.1–95.0%)	ETEC VPO + TD severity ≥ 5	ETEC VPO (alt) and TD severity score ≥ 5
57.7% (−9.5–83.6%)	MS-ETEC VPO	≥3 unformed stools in 24 h, with moderate to severe abdominal pain/cramps, nausea, or vomiting; or change in activity due to illness plus ETEC detected at any time during disease
54.4% (−19.2–82.1%)	MS-ETEC VPO + TD severity ≥ 3	MS-ETEC VPO and TD severity score ≥ 3
83.5% (26.7–96.3)	MS-ETEC VPO + TD severity ≥ 4	MS-ETEC VPO and TD severity score ≥ 4
87.7% (1.6–98.5%)	MS-ETEC VPO + TD severity ≥ 5	MS-ETEC VPO and TD severity score ≥ 5

**Table 8 microorganisms-11-02414-t008:** Anti-CTB serum IgA titer in US Travelers (N = 646) Upon Arrival in Guatemala or Mexico, and the Relative Risk (RR) of ETEC Strain Infection and Development of MS-ETEC TD.

ETEC Strain	Arrival Anti-CTB IgA Titer	Number of Participants	Total Infections (%)	TD ^2^	MS-ETEC TD ^3^
LT, ST or LT/ST ^1^	≤360	383	40 (10.4%)	35 (8.6%) Median Duration = 6 days	18 (4.7%) Median Duration = 6 days
	>360	263	10 (3.8%)	7 (2.6%) Median Duration = 2.5 days	1 (0.4%) Median Duration = 1.5 days
			RR = 0.26 (0.14–0.35; *p* = 0.002)	RR = 0.24 (0.18–0.38; *p* = 0.001)	RR = 0.37 (0.29–0.44; *p* = 0.001)
ST ^4^	≤360	383	40 (10.4%)	36 (9.4%) Median Duration = 4 days	28 (7.3%) Median Duration = 4 days
	>360	263	38 (14.4%)	31 (11.8%) Median Duration = 6 days	28 (10.6%) Median Duration = 5 days
			RR = NS	RR = NS	RR = NS

^1^ LT, ST, or LT/ST ETEC as sole pathogen isolated at any time during the TD episode and sharing an antigen with the vaccine. ^2^ >3 loose or watery stools in a 24-h period, accompanied by abdominal pain or cramps, nausea, or vomiting of any intensity, plus ETEC as the sole pathogen isolated. ^3^ ETEC TD plus at least one moderate to severe GI symptom or ETEC TD, or changes in activity due to illness severity. NS = Not Significant. ^4^ Includes strains that did not have a detectable CF/CS antigen, as well as those that expressed a CF/CS antigen that was not present in the vaccine (i.e., CS6).

**Table 9 microorganisms-11-02414-t009:** Anti-CTB serum IgA titer in US Travelers (N = 646) Upon Arrival in Guatemala or Mexico, and the Relative Risk (RR) of Infection with Other Enteric Pathogens and Development of Moderate to Severe-TD.

Enteric Pathogen	Arrival Anti-CTB IgA Titer	Number of Participants	Total Infections (%)	TD ^1^	MS-ETEC TD ^2^
*Campylobacter jejuni/E. coli* ^3^	≤360	383	17 (4.4%)	9 (2.3%) Median Duration = 6 days	9 (2.3%) Median Duration = 6 days
	>360	263	4 (1.5%)	3 (1.1%) Median Duration = 7 days	3 (1.1%) Median Duration = 7 days
			RR = 0.34 (0.12–1.1; *p* = 0.03)	RR = 0.48 (0.13–1.8; *p* = NS)	RR = 0.48 (0.13–1.8; *p* = NS)
*Salmonella* spp. ^4^	≤360	383	9 (2.3%)	3 (0.8%) Median Duration = 7 days	3 (0.8%) Median Duration = 7 days
	>360	263	2 (0.8%)	1 (0.4%) Median Duration = 1 day	1 (0.4%) Median Duration = 1 day
			RR = 0.32 (0.07–1.5; *p* = NS)	RR = 0.48 (0.05–4.6; *p* = NS)	RR = 0.48 (0.05–4.6; *p* = NS)
*Shigella* spp.	≤360	383	5 (1.3%)	5 (1.3%) Median Duration = 6 days	5 (1.3%) Median Duration = 6 days
	>360	263	6 (2.3%)	6 (2.3%) Median Duration = 6 days	5 (1.3%) Median Duration = 6 days
			RR = NS	RR = NS	RR = NS

^1^ >3 loose or watery stools in a 24-h period, accompanied by abdominal pain or cramps, nausea, or vomiting of any intensity, plus ETEC as the sole pathogen isolated. ^2^ ETEC TD, plus at least one moderate to severe GI symptom or ETEC TD, or changes in activity due to illness severity. ^3^ At a higher arrival threshold, anti-CTB serum IgA titer of ≥640, OKV significantly reduced the risk of *Campylobacter* infection, in addition to protection against *Campylobacter* TD (RR = 0.29, CI 0.06–1.3; *p* = 0.05) and *Campylobacter* MS-ETEC TD (RR = 0.13, CI 0.02–1.0; *p* = 0.02). LT, ST, or LT/ST ETEC as sole pathogen isolated at any time during the TD episode. ^4^ A higher arrival threshold anti-CTB serum IgA titer of ≥640 does not make a difference in a reduced risk of *Salmonella* infection, TD, or MS-ETEC TD. NS = Not Significant.

## Data Availability

Data are contained within the article and in associated Appendix A. The article will be published in a fully open access journal to help ensure widespread data dissemination.

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
