# Peer review of "Efficacy of an Enterotoxigenic Escherichia coli (ETEC) Vaccine on the Incidence and Severity of Traveler’s Diarrhea (TD): Evaluation of Alternative Endpoints and a TD Severity Score"

_microorganisms, 2023, doi:10.3390/microorganisms11102414_

Round 1

Reviewer 1 Report

This is an important and well-written manuscript that reports the post hoc reanalysis of study data from an earlier clinical trial of an oral whole-cell killed ETEC vaccine conducted in travelers to endemic regions in Mexico and Guatemala.  The original study employed a strict endpoint of vaccine preventable ETEC travelers diarrhea that relied on ≥ 5 unformed stools in a 24 hour period while in the reanalysis more liberal definitions were included. The was no protection afforded by the vaccine based on the original endpoint although relatively few subjects (13 altogether) actually met the original definition. Notably however there was significant protection (PE 76.3 %) against moderate-severe ETEC diarrhea, and against higher TD severity score in 251 vaccinees who met the definition of vaccine take (serum IgA to CTB titer >360) on arrival. This parameter was important given the self-adminstration of the vaccine by the study subjects. Overall the results are likely to impact the design of similar studies in the future. 

I have only some minor corrections for the authors to consider in redrafting their manuscript. 

minor formatting/grammatical issues:

1. line 58 affecting not effecting

2. period after buffer line 102.

2. reformat table 1 as in present form definitions run together. 

3. line 255 and table S3. vomiting post-dose was higher in recipients vs placebo controls although which of these differences were significant is confusing as written. 

4. table S3 last line subject not subjects. 

5. line 441/table 8 it might be good to point out that the infections column refers to asymptomatic infection? ?

Author Response

Response to Reviewer 1 Comments

Thank you very much for taking the time to review this manuscript. Please find the detailed responses below and the corresponding revisions/corrections highlighted in track changes in the re-submitted files.

Comment 1: line 58 affecting not effecting

Response 1: Thank you, this has been corrected.

Comment 2: period after buffer line 102.

Response 2: Thank you, this has been corrected.

Comment 3: reformat table 1 as in present form definitions run together. 

Response 3: Thank you for the comment; Table 1 has been reformatted per reviewer suggestion.

Comment 4: line 255 and table S3. vomiting post-dose was higher in recipients vs placebo controls although which of these differences were significant is confusing as written. 

Response 4: Thank you for pointing out the confusion; we have further clarified the text in line 255 and Table S3.

Comment 4: table S3 last line subject not subjects.

Response 4: Thank you, this has been corrected.

Comment 5: line 441/table 8 it might be good to point out that the infections column refers to asymptomatic infection? ?

Response 5: Table 8 refers to total infections (symptomatic and asymptomatic) and has been updated to reflect this point. Additionally, the reference to ‘asymptomatic’ was removed from line 441 for further clarification.

Reviewer 2 Report

Peer review: manuscript microorganisms-2582206 ("Efficacy of an ETEC Vaccine on the Incidence and Severity of Traveler’s Diarrhea (TD): Evaluation of Alternative Endpoints and a TD Severity Score”).

Comments to the Authors:

The research question is very important. Travelers’ diarrhea (TD) is one of the most common travel-related illness. As described by the authors, the study evaluated the role of vaccine immune responses on vaccine efficacy and calculated efficacy using novel considerations regarding the impact of the TD on activities as well as the novel disease severity score. The whole study seems well-designed and interesting results are presented. However, the main results were obtained after a reanalysis of data from a Phase III double-blind, randomized, placebo-controlled vaccine efficacy field trial [Sack et al., 2007]. Now the authors used alternative disease endpoints and the TD severity score which capture a wider spectrum of illness severity as compared to the original study endpoint. Some previous data were reported to be presented in a Congress in 2007 [Burgeois et al, 2007].

My opinion is that the authors have to clarify the motivation for re-analyzing data from a study carried out more than 15 years ago. It is also necessary to provide the Ethics Committee approval number (even if it is for a data re-analysis). I consider this information essential before a more meticulous review of the article.

Another some suggestions before the re-submission:

1)    Avoid abbreviation (Ex.: ETEC) in the Title.

2)    Prepare better the Tables and Figures (Ex.: remove letters font in red).

3)    Why was calculated protection and not risk in all evaluations? I think it would be more interesting to present the risk (instead protection).

It seems OK. 

Author Response

Response to Reviewer 2 Comments

Thank you very much for taking the time to review this manuscript. Please find the detailed responses below and the corresponding revisions/corrections highlighted in track changes in the re-submitted files.

Comment 1: Avoid abbreviation (Ex.: ETEC) in the Title.

Response 1: Thank you, this has been corrected.

Comment 2: Prepare better the Tables and Figures (Ex.: remove letters font in red).

Response 2: Thank you; all tables have been reformatted per reviewer suggestion.  

Comment 3: Why was calculated protection and not risk in all evaluations? I think it would be more interesting to present the risk (instead protection).

Response 3: Thank you for the comment. This study was originally designed to assess the protective efficacy (PE) of a novel inactivated ETEC vaccine. In keeping with the same context as the primary objectives of the research, the ability of the vaccine to reduce the relative risk of diseases is reported in the same manner as was originally planned in the conduct of the trial – i.e., as protective efficacy. Relative risk remains estimable from the data provided in the tables.

Additional Clarifications:

My opinion is that the authors have to clarify the motivation for re-analyzing data from a study carried out more than 15 years ago.
Please see lines 77-79 where it explains rationale for application of the score to the OEV-118 dataset. Nevertheless, additional rationale was also added to the discussion section (lines 421-423).

It is also necessary to provide the Ethics Committee approval number (even if it is for a data re-analysis). I consider this information essential before a more meticulous review of the article.
Thank you for the comment. The Joint Committee on Clinical Investigation at Johns Hopkins University tracked their protocol submissions and approvals via the Protocol number, which has now been added to the manuscript along with the date of initial approval (line 113).

Round 2

Reviewer 2 Report

The authors made the main suggested changes and answered the questions I raised previously. However, the title still has an abbreviation (ETEC). My opinion is that it is important to inform that it is a vaccine for enterotoxigenic Escherichia coli (and not just put ETEC).

Furthermore, the Abstract text will be used to widely publicize the article. Therefore, I recommend additional efforts from the authors focused on improving this text to allow an understanding of the entire study after reading it. Mainly, I suggest including one or two sentences of introduction to the main topic of the study at the beginning, as well as avoiding excessive detailing of technical procedures, which can only be described in the manuscript (Ex.: " An immunologic take was defined as anti-CTB serum IgA titer >360. " )

 Moderate editing of English language required
